

# Root fragment weight and carbohydrate dynamics of two weedy thistles *Cirsium arvense* (L.) Scop. and *Sonchus arvensis* L. during sprouting

Marian Malte Weigel[1,2], Sabine Andert[1,2], Manuela Alt[3], Kirsten Weiß[3], Jürgen Müller[4] and Bärbel Gerowitt[2]

[1] Julius Kühn Institute (JKI)–Federal Research Centre for Cultivated Plants, Institute for Plant Protection in Field Crops and Grassland, Braunschweig, Germany
[2] Faculty of Agricultural and Environmental Sciences, Crop Health, University of Rostock, Rostock, Germany
[3] Albrecht Daniel Thaer-Institute of Agricultural and Horticultural Sciences, Humboldt University of Berlin, Berlin, Germany
[4] Faculty of Agricultural and Environmental Sciences, Group Grassland and Forage Science, University of Rostock, Rostock, Germany

Corresponding author
Marian Malte Weigel,
marianmalteweigel@gmail.com

## ABSTRACT

Understanding the carbohydrate dynamics of sprouting *Cirsium arvense* (L.) Scop. and *Sonchus arvensis* L. ramets can assist in optimizing perennial weed management. However, detailed knowledge about general reserve dynamics, minimum values in reserves (compensation point) and different reserve determination methods remains sparse. We present novel insights into reserve dynamics, which are especially lacking for *S. arvensis*. We uniquely compare root weight changes as a proxy for carbohydrates with direct carbohydrate concentration measurements using high-performance liquid chromatography (HPLC). In a greenhouse study, ramets of two sizes (20 and 10 cm) were planted in pots. Subsequent creeping roots of sprouted plants were destructively harvested and analyzed for carbohydrates 12 times between planting and flowering. Efficiency in storing carbohydrates and the replenishing rate of root weight and carbohydrates was much higher in *S. arvensis* than in *C. arvense*. Thus, our study urges to evaluate perennial weed species individually when investigating root reserves. Determining root reserves by either using root weight changes as a proxy for carbohydrates or directly measuring carbohydrate concentrations by HPLC differed in the minimum values of reserves referred to as compensation points. For both species, these minimum values occurred earlier based on root weight than based on carbohydrate concentrations. Cutting ramets into 20 or 10 cm sizes did not significantly affect carbohydrate concentration or root weight changes for both species. We conclude that any practical applications targeting perennial weeds by fragmenting roots into small ramets through belowground mechanical control must be evaluated for trade-offs in soil structure, soil erosion, and energy consumption.

## INTRODUCTION

*Cirsium arvense* (L.) Scop (Creeping thistle) and *Sonchus arvensis* L. (Perennial sow-thistle) are two creeping perennial weed species from the Asteraceae plant family. Both can cope with arable conditions, meaning plants thrive on sites with regular soil disturbance, periods with strong competition by annual crops alternating with those of low competition (*Vanhala, Lötjönen & Hurme, 2006*; *Favrelière et al., 2020*). The two species are described to occur under temperate conditions. While *C. arvense* is frequently researched under arable conditions worldwide (*Wilson, Martin & Kachman, 2006*; *Rodriguez et al., 2007*; *Verwijst et al., 2013*) there are fewer on *S. arvensis* (*Tørresen et al., 2022*). The latter species is mainly studied under northern European conditions (*Vanhala, Lötjönen & Hurme, 2006*; *Liew et al., 2012*; *Anbari et al., 2016a*).

Both species produce creeping roots. While nutritious roots are primarily responsible for the uptake of water and nutrients from the soil, the task of creeping or adventitious roots is vegetative expansion and dispersal. Both species expand horizontally with these creeping roots and produce new shoots from clonal growth (Fig. 1, Fig. A1). For dispersal, the special feature of creeping roots to propagate from fragments is of fundamental importance (*Nadeau & Born, 1989*; *Lemna & Messersmith, 1990*). These root fragments are propagules as they can grow into a new plant after being detached from the rest. In the realm of population biology, each seed represents a genet, hence a genetically different organism, whereas fragments resulting from clonal growth exhibit genetically identical ramets (*Harper, 1977*). Therefore, creeping roots that have been fragmented prior to new sprouting can be called ramets. Comparable to seeds, ramets enable early growth of the new plant from their reserves. Ramet sprouting and the establishment of new plants starts with a heterotrophic phase in which the new shoots rely on the carbohydrates stored in the ramet.

Both thistle species ensure their lifeform by these carbohydrate storing creeping roots (*Pegtel, 1973*; *Tworkoski, 1992*). The function of those in surviving phases without assimilate supply and in sprouting again have been of fundamental interest to weed scientists for a long time (*Arny, 1932*). As reserves are depleted, minimum values in creeping root reserves during growth and ontogenetic development are expected (Fig. 1). One particular point of interest and major subject of several studies on *C. arvense* is referred to as the compensation point (*Håkansson, 2003*; *Rodriguez et al., 2007*; *Nkurunziza & Streibig, 2011*; *Verwijst et al., 2013*).

*Håkansson (2003)* defines the compensation point as the point where the amount of new photo-assimilated carbon exceeds the amount of carbon being allocated from stored resources. Thus, the compensation point is reached when newly established shoots become self-sufficient from their previous dependency on root carbohydrates (*Håkansson, 2003*). Proper identification of the compensation point is vitally important, as

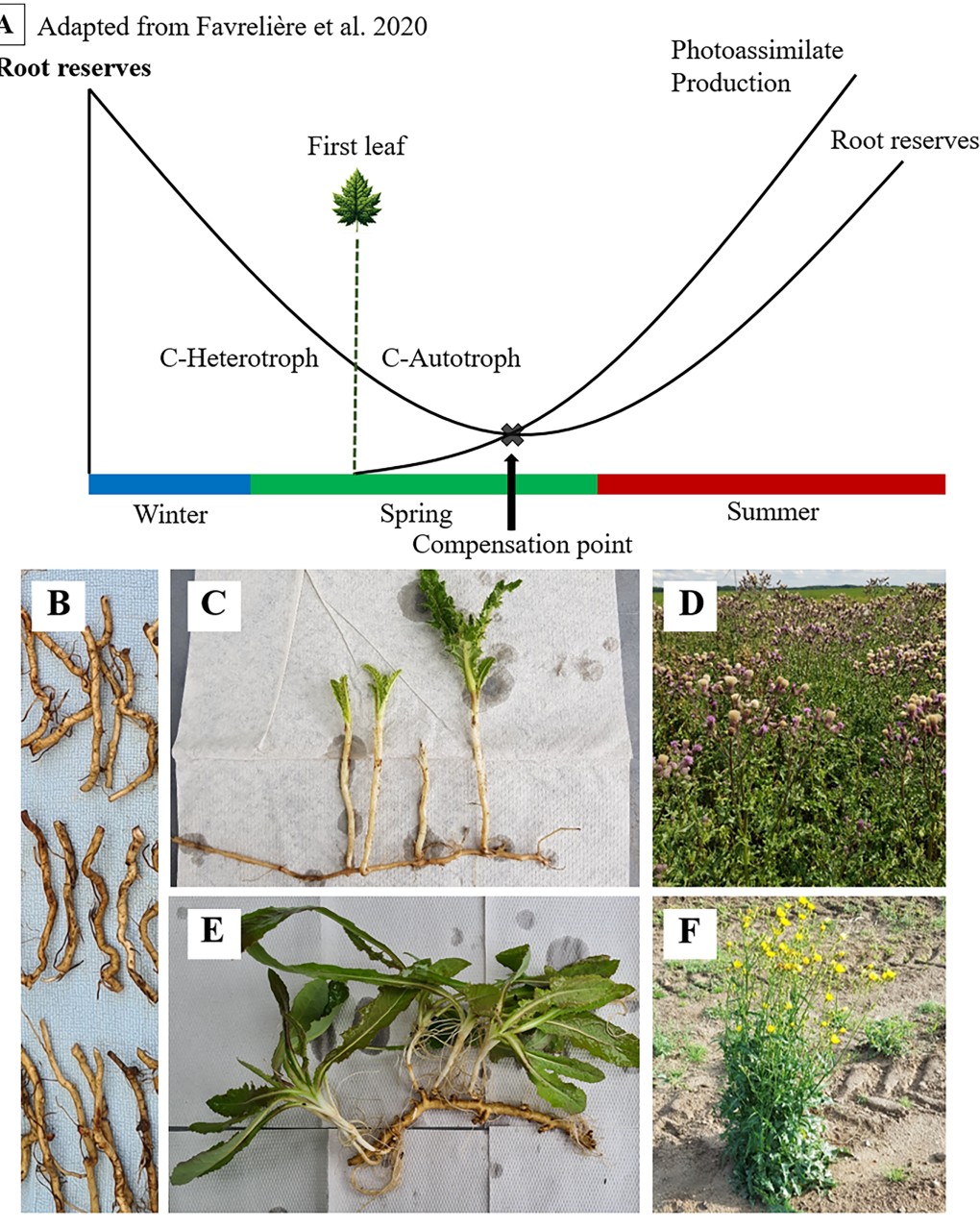

**Figure 1  Seasonal dynamics of root reserves and photoassimilate production (A) of *C. arvense* (C, D) and *S. arvensis* (E, F).** Starting as an overwintering ramet (B), sprouting, transitioning from C-heterotrophy in winter to C-autotrophy post "First leaf" emergence in spring and reaching the compensation point (C, E), followed by reserve replenishment till reaching maximum reserve accumulation after the onset of flowering (D, F).

the weakest resprouting capabilities of ramets align with minimum points in reserves (*Dock Gustavsson, 1997*; *Håkansson, 2003*). For this very reason, both mechanical and chemical control methods are directed towards this point (*Håkansson, 2003*; *Nkurunziza, 2010*).

The compensation point for *C. arvense* was identified to coincide between three (*Verwijst, Tavaziva & Lundkvist, 2018*) and 12 leaves (*Rodriguez et al., 2007*). In a series of recent studies, the carbohydrate amount was derived using root weight as a proxy, assuming correlations between root weight and the carbohydrate amount stored in these propagules (*Dock Gustavsson, 1997*; *Håkansson, 2003*; *Verwijst, Tavaziva & Lundkvist, 2018*). Assuming correlations between root weight and the carbohydrate amount, these compensation points represent minimum values in the amount of stored reserves. However, others directly evaluated the carbohydrate concentrations through HPLC measuring (*Rodriguez et al., 2007*; *Nkurunziza & Streibig, 2011*). Hence, the point in time at which the compensation point is reached may vary based on the determination method by either using root weights as a proxy for reserve amount or carbohydrate concentrations. Studies investigating the compensation point were exclusively conducted on *C. arvense*. It is not clear whether the connections between root weight, carbohydrate concentration and their corresponding compensation points can be generalized for species belonging to the Asteraceae plant family.

In arable farming, creeping perennials are managed, either chemically or non-chemically (*Favrelière et al., 2020*). Popular and effective for non-chemical weeding is to disturb the roots by ploughing (*Thomsen, Brandsæter & Fykse, 2013*; *Brandsæter et al., 2017*). As creeping perennials are known to be sensitive to belowground disturbance, this is a suitable method to manage them (*Brandsæter et al., 2017*). Through ploughing, creeping roots are fragmented into ramets and buried in the soil (*Vanhala & Salonen, 2007*). Ploughing stimulates resprouting, however, it may result in new shoots that deplete root reserves (*Håkansson, 2003*). Ramet size was identified to be an important factor in influencing carbohydrate dynamics as the relative performance of planted roots was shown to be attributed to the amount of stored carbohydrates (*Dock Gustavsson, 1997*; *Thomsen, Brandsæter & Fykse, 2013*; *Verwijst et al., 2013*). Hence, it seems crucial for any success of control that ramets are fragmented at the right time and into the right size.

In this study we evaluated the root weight and the carbohydrate concentration of creeping roots originating from ramets. Combined, these two variables give the total amount of carbohydrates. Ramets of *C. arvense* and *S. arvensis* species were included in two sizes. At each harvest date, a subset of plants was destructively harvested, allowing for a sequential assessment of root weight and carbohydrate concentration from planting until flowering in both species.

This study included two species of the same family, *C. arvense* and *S. arvensis*, sharing the same life-form, to test the hypothesis (1) that carbohydrate dynamics in the roots of the related perennial species follow a similar pattern.

With more reserves at planting time, larger ramets can grow longer solely from these reserves than smaller ramets. Hence, we hypothesize (2) that large ramets reach the compensation point later than the smaller ones.

Two methods to determine root reserves were applied in previous studies, either using root weights as a proxy for reserves or directly measuring carbohydrates through HPLC analysis. We hypothesize (3) that the occurrence of minimum values in root reserves over time depends on the method used.

## MATERIALS AND METHODS

We conducted two greenhouse pot experiments, one in winter 2020/21 and one in spring 2021 in Rostock, Germany.

### Plant material

Root material in the form of ramets of *C. arvense* was obtained from an experimental field in Rostock, Germany (54°03′39.5″N 12°05′03.9″E), in November of 2020 for the first and in March 2021 for the second run of the experiment. The field had been left as fallow land since 2018.

Ramets of *S. arvensis* were taken from a root bank kept in Rostock. Its root material had originally been collected from an organically managed farm close to Rostock (53°47′37.15″ N 12°10′33.5″E). New root material was then propagated from this collected material in outdoor pots. After extraction, root material of both species was stored dark at +2 °C for approximately 2 months. Root material was stored and planted in a soil mix of arable soil (sandy loam), garden mold, and compost in a 2:1:1 ratio with a pH of 5.7.

### Experimental set up

One day before planting the roots were cut into pieces (Table 1). Ramets of both species were sorted into two groups with five ramets each differing in ramet sizes (length and weight) (Size–L = Large and Size–S = Small) (Table 1). In our study, the term ramet size is defined by a combination of weight and length. While we precisely measured the initial ramet weights, there was a small range of variation in length. The cumulated total weight (fresh weight) of the ramets for each species doubled while the length approximately doubled (Table 1). After preparing the ramets, each one was planted in a separate pot. The pot volume was 10 liters with a surface area of 0.07 m$^2$. Planting depth of ramets was 10 cm. The depth of the roots in the fields was observed when digging-up pieces in previous experiments. Based on these observations we chose a depth of 10 cm to represent the field depth of the horizontally creeping roots as best as possible. After planting, the pots were irrigated and kept moist during the whole experimental period.

After planting, pots were placed inside the greenhouse following a fully randomized design. This methodology of preparing and planting ramets was consistent across both experiments.

Each group of species and ramet size was established 13 times, resulting in 65 plants per species and size and 260 plants in total. Five pots for each group were assessed at every destructive sampling. After planting the ramets, each group was harvested 12 times according to a distinct accumulated temperature sum given as:

$$GDD = ([T_{max} + T_{min}]/2) - T_{base}. \tag{1}$$

*McMaster & Wilhelm (1997)* and *Donald (2000)* suggested to cumulate growing degree days (GDD) after April 1 (day 91 of the year) above a base temperature ($T_{base}$) of 0 °C when predicting emergence dates of perennial weeds. This date is intended to prevent temperatures below 0 °C from affecting the accuracy of the model while simultaneously considering the growth patterns of both species. In order to best simulate outdoor

**Table 1 Measurements of ramets at time of planting.**

| Species | Size | Ramet diameter Ø | Ramet length | Fresh weight five ramets |
|---|---|---|---|---|
| *C. arvense* | Small | 4–5 mm | 10–12 cm | 13 g |
| *C. arvense* | Large | 4–5 mm | 20–22 cm | 26 g |
| *S. arvensis* | Small | 5–6 mm | 10–12 cm | 14 g |
| *S. arvensis* | Large | 5–6 mm | 20–22 cm | 28 g |

Note:
Ramets at time of planting divided into four groups (species × size) of five pieces each.

conditions, we mimicked the temperatures and photoperiods of Rostock, Germany, in the greenhouse. These temperatures recorded in the greenhouse were used for the following calculations. The calculation was started with day 92 of the year 2019, based on information provided by a nearby weather station (DWD, Station id 04270), (Table A1). The greenhouse was equipped with supplementary light sources in order to simulate similar light levels to field conditions (Table A2). Harvesting pots began at 200 GDD, approximately 21 days after planting. We harvested the pots every 100 GDD days following the initial harvest up to the final harvest day at 1,300 GDD days. Both species were flowering at the 12 and thus latest date. The factor "Experimental time" carries these 12 sequential harvests, always measured as cumulated GDD.

## Assessments

Plants were always evaluated individually at each harvest date. For aboveground plant parts biomass and number of leaves was measured. According to *Verwijst, Tavaziva & Lundkvist (2018)* we counted the leaves of the shoot with the highest number of leaves. We refer to this as the most developed shoot. The belowground plant parts were divided into belowground shoot parts, adventitious roots and nutritious roots. The evaluations and analyzes included all adventitious root parts, meaning originally planted and newly formed parts (Fig. A1). Nutritious roots and belowground shoot parts were not taken into account for any of the following evaluations.

Assessments were:

– Root weight: Dry weight of all belowground adventitious roots in g DM
– Number of leaves: Number of leaves of the most developed shoot per plant
– Days after emergence (DAE): Days from emergence of first shoot until harvest date
– Days till emergence (DTE): Days until emergence of the first shoot.

## Carbohydrate content

All adventitious root parts of one group were grinded and homogenized by a laboratory rotary mill (Brabender, Duisburg, Germany) equipped with a mesh wide of 1 mm before being analyzed for their carbohydrate content (RCH). We regarded the sum of single sugars and inulin-like fructans as root carbohydrate reserves giving the concentration of root carbohydrates (RCH concentration). Their contents were determined by high performance liquid chromatography (HPLC) in aqueous extracts. Extracts were prepared

by blending 500 mg (±20 mg) of the ground, air-dried sample material with 100 ml ultrapure water (TOC-free). The 250 ml-bottles were tightly closed, shaken for 1 h with medium speed at room temperature. The solutions were filtered through a 125 mm diameter folded filter (Whatman 595 ½) by micro-filtration (Minisart CHROMAFIL Xtra RC-45/25) into a 2 ml HPLC vial before starting the measurement. Single sugars and fructan (as inulin) were detected by HPLC. The HPLC system (LC 20; Shimadzu, Kyoto, Japan) was composed of an autosampler with a storage temperature of 4 °C, a column oven (85 °C), and a refraction index detector (RID). The measurement run with flow rate of 0.4 ml/min was isocratic with column Nucleosil CHO 682 (Pb Machery-Nagel) with a length of 300 mm × 7.8 mm column diameter, combined with a precolumn cartridge (21 mm × 4.6 mm). A mobile phase HPLC-grade water was used, and the injection volume was 20 µl. The limit of detection for each parameter was determined with 1 mg/l. Additional information on the composition and preparation of the single sugars series used in our experiments can be found in *Weiß & Alt (2017)*.

## Statistical analysis

Data were analyzed using a linear mixed-effects model in the statistics environment R (R Core Team, Vienna, Austria; version 3.6.3, 2020); package "lme4" (*Bates et al., 2015*). The fixed effects included in the model were thistle species, ramet size, experimental time (given by the 12 harvest dates), and the main interactions between these effects. Random effect were the two repetitions of the experiment.

The samples of the five replicates were pooled to obtain enough material for reliable carbohydrate measurements. To account for this missing of true replicates a Type III analysis of variance (ANOVA) was conducted, with the Satterthwaite method applied to estimate degrees of freedom. This approach, implemented in the R- package "lmerTest", allows to accurate capture the variance sources between the groups (*Satterthwaite, 1946*; *Kuznetsova, Brockhoff & Christensen, 2017*). The square of the correlation ratio (eta squared, $\eta^2$) estimates the effect sizes of the fixed effects. Experimental time was treated as covariate in the statistical model. Due to the non-linear response over time, a second-degree 2$^{nd}$-degree polynomial regression model was selected as it effectively represented the non-linear interactions between experimental time, species and ramet size. Given the observed curvilinear response of root weight and carbohydrate concentration over time, this model provided a more accurate fit than classical growth or degradation models, capturing the gradual and complex nature of root reserve dynamics.

Score data were analyzed for factorial effects using the Kruskal-Wallis $\chi^2$ test. Throughout this study, we regard *p*-values < 0.05 as statistically significant.

## RESULTS

The response variables root weight and carbohydrate content were analyzed with respect to the experimental factor species, ramet size and experimental time. These data allow to model root weight and reserves as a function of thermal time for both species.

## Root weight

The analysis of variance revealed a significant effect of the experimental time on the root weight (Table 2). Expression of ramet weight proved largely independent of the thistle species, showing only a very low effect size (partial $\mu^2$) and therefore low potential to be a source of root weight variance. While ramet size was found to significantly affect root weight ($p = 0.0025$), this effect was largely attributed to the experimental design, where larger ramets inherently had more root weight due to their initial size (Tables 1, 2).

Attention should, therefore, be drawn to the significant interaction 'Thistle species × Experimental time', stating that the effects of experimental time on the root weight cannot be considered independently of the studied species. Consequently, we segregated the two thistle species for all subsequent analyses.

However, the random effect of the trial series on the test characteristic 'root weight' was not significant ($\chi^2$-Test, $p > 0.05$). Therefore, the two trial repetitions were considered as replicates in the further data analysis.

## Root carbohydrates

Thistle type significantly influenced root carbohydrates (RCH) concentration (Table 3). The interaction 'Thistle species × Experimental time' had the highest explanatory power among the putative influencing factors investigated in the experiment (see partial $\mu^2$ as an effect size estimate in Table 3). By enclosing both species based on the analysis of variance it is worth mentioning that experimental time alone proved to be of no verifiable impact. Neither the size of the buried ramet nor its interaction with the two different thistle species revealed a formative influence on the RCH concentration.

An analysis of variance was also carried out for the derived characteristic 'Amount of RCH', being the product of root weight with RCH concentration (Table 4). As in the ANOVA for the target trait root weight, experimental time had the strongest effect, followed by the significant interaction 'Thistle species × Experimental time'.

## Modelling root reserves over time

The courses of dependent variables characterizing the reserves of roots for re-sprouting after their burial followed a nonlinear curve. Hence, 2nd degree polynomial regression models were chosen to represent the trajectories as estimating equations (Fig. 2). Due to the pronounced interaction 'Thistle species × Experimental time', modelling of trends is carried out separately for each species for the two response variables root weight and RCH concentration.

Modelled curve of root weight development of *C. arvense* after burial is shown in Fig. 2A. In order to make the fitted model visible, evaluated data was plotted in addition to the shown trend line. The appearance of the first leaf, therefore, ended the exclusively heterotrophic phase of reserve metabolism. The minimum point of the curve (compensation point) is reached 148 GDD days after the appearance of the first leaf. Thus, for the roots to regain the ramet weight at the time of planting required 800 GDD.

The RCH concentration of *C. arvense* roots (Fig. 2B) reached its minimum 389 GDD days later than minimum of root weight (Fig. 2A). Time needed to reach the compensation

**Table 2 Effects of the factors thistle species, ramet size, experimental time, and their interactions on the root weight.**

| Source of variation | Mean square | Num DF/Den DF | F-value | p-value | Partial μ² |
|---|---|---|---|---|---|
| Thistle species | 30.7 | 1/100.03 | 0.974 | 0.3260 NS | 0.01 |
| Ramet size | 302.1 | 1/99.99 | 9.577 | 0.0025** | 0.09 |
| Experimental time | 6,590.2 | 1/100.11 | 208.906 | <0.0001*** | 0.68 |
| Thistle species × Experimental time | 462.2 | 1/100.11 | 14.651 | 0.0002*** | 0.13 |
| Thistle species × Ramet size | 0.1 | 1/99.99 | 0.002 | 0.9673 NS | 0.00 |

Note:
Presented are the results of Type III Analysis of Variance with Satterthwaite's method for estimation the degrees of freedom (DF). Num DF, Numerator DF; Den DF, Denominator DF; NS, not significant; partial μ-effect size.
** $p < 0.01$.
*** $p < 0.001$.

**Table 3 Effects of the factors thistle species, ramet size, experimental time, and their interactions on RCH concentration of the buried roots.**

| Source of variation | Mean square | Den DF/Num DF | F-value | p-value | Partial μ² |
|---|---|---|---|---|---|
| Thistle species | 199,952 | 1/102 | 13.549 | 0.0004*** | 0.12 |
| Ramet size | 19,052 | 1/102 | 1.291 | 0.2585 NS | 0.01 |
| Experimental time | 25,266 | 1/102 | 1.712 | 0.1937 NS | 0.02 |
| Thistle species × Experimental time | 509,022 | 1/102 | 34.491 | <0.0001*** | 0.25 |
| Thistle species × Ramet size | 38,625 | 1/102 | 2.617 | 0.1088 NS | 0.03 |

Note:
Presented are the results of Type III Analysis of Variance with Satterthwaite's method for estimation the degrees of freedom (DF). Num DF, Numerator DF; Den DF, Denominator DF; NS, not significant; partial μ-effect size.
*** $p < 0.001$.

**Table 4 Effects of thistle species, ramet size, experimental time, and their interactions on RCH amount of the buried roots.**

| Source of variation | Mean square | Den DF/Num DF | F-value | p-value | Partial μ² |
|---|---|---|---|---|---|
| Thistle species | 146.49 | 1/100 | 6.903 | 0.010** | 0.06 |
| Ramet size | 81.05 | 1/100 | 3.819 | 0.053 NS | 0.04 |
| Experimental time | 2,435.89 | 1/100 | 114.793 | <0.0001*** | 0.53 |
| Thistle species × Experimental time | 1,490.68 | 1/100 | 70.249 | <0.0001*** | 0.29 |
| Thistle species × Ramet size | 29.51 | 1/100 | 1.391 | 0.2411 NS | 0.01 |

Note:
Presented are the results of Type III Analysis of Variance with Satterthwaite's method for estimation the degrees of freedom (DF). Num DF, Numerator DF; Den DF, Denominator DF; NS, not significant; partial μ-effect size.
** $p < 0.01$.
*** $p < 0.001$.

point by RCH concentration (Fig. 2B) corresponded roughly to the time period roots needed to regain the starting ramet weight (Fig. 2A). After the appearance of the first leaf, it took another 537 GDD days until the RCH concentration reached its absolute minimum point ($CP_{CH}$ in Fig. 2B).

Both species root weights developed in a similar pattern, but *S. arvensis* was faster (Figs. 2A, 3A). According to root weight ($CP_{RW}$), there was a time gap of 90 GDD days between the beginning of photosynthesis as indicated by the occurrence of the first leaf and the compensation point. In contrast to *C. arvense*, the weight progression curve of buried *S. arvensis* roots still fits well after more than 1,000 GDD days.

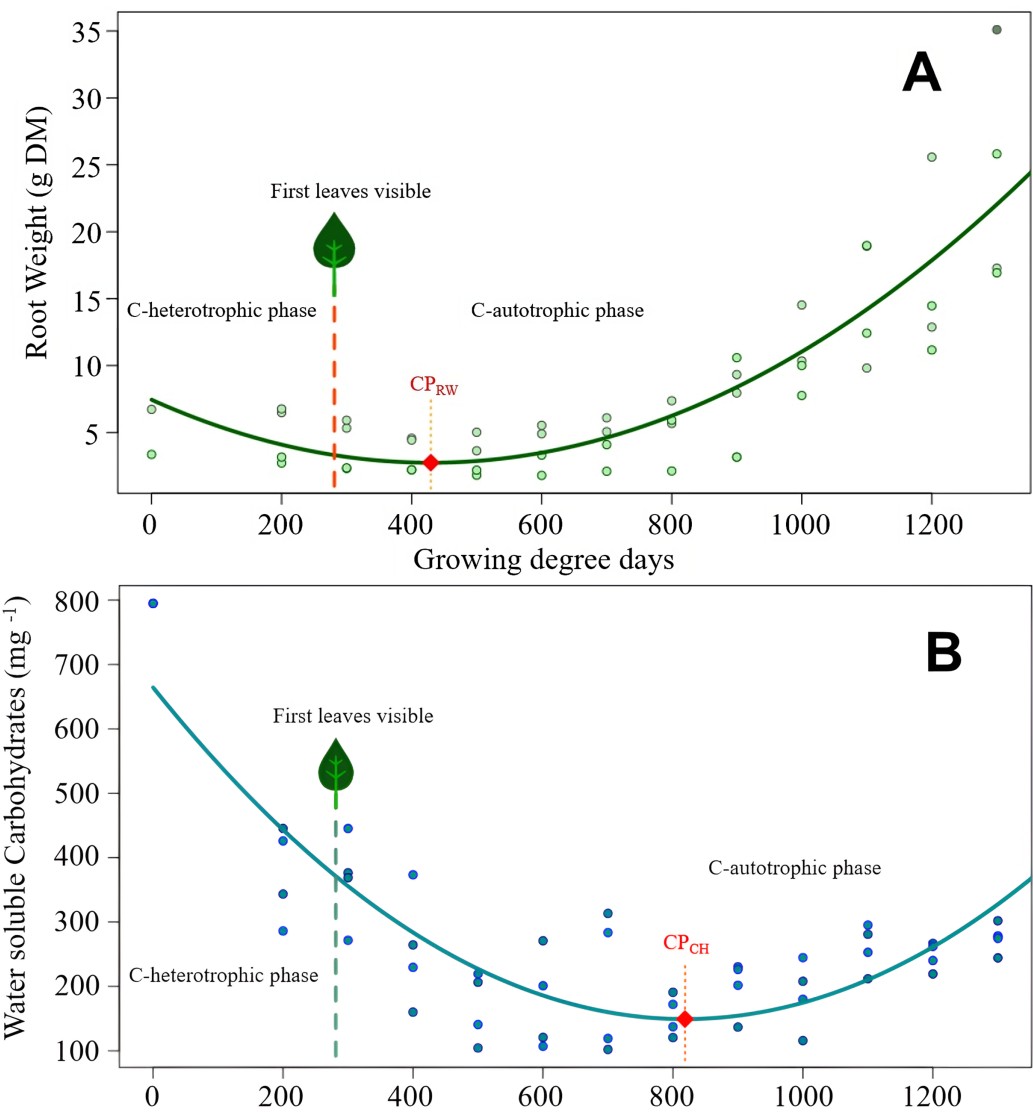

**Figure 2** *Cirsium arvense* **root reserves over time.** (A) Root weight development of *C. arvense* as a function of experimental time; RW = 7.45 − 0.0219GDD + 0.0000255GDD2, $R^2$ = 0.75***. Model regression line (in green) and original data (as dots). The dashed line with a leaf at the top indicates the time of first leaf appearance, representing the shift from the C-heterotrophic to the C-autotrophic phase. The red square on the regression line marks the minimum (compensation point) according to ramet weight (CPRW) in g DM. (B) Development of RCH concentration as water soluble carbohydrates (WSC) in ramets of *C. arvense* as a function of experimental time. WSCH = 664.4 − 1.259 GDD + 0.000769 GDD2, $R^2$ = 0.73***. Model regression line (in turquoise) and the original data (as dots). The dashed line with a leaf at the top indicates the time of first leaf appearance, representing the shift from the C-heterotrophic to the C-autotrophic phase. The red square on the regression line marks the minimum (compensation point) according to the RCH concentration (CPCH).

The compensation point according to the root RCH (CP$_{CH}$) of *S. arvensis* (Fig. 3B) was reached 288.5 GDD days earlier than for *C. arvense* (Fig. 2B). Moreover, the RCH concentration balance of *S. arvensis* recovered faster from the heterotrophic phase than that of *C. arvense* (Figs. 2B, 3B). The scatter of RCH contents along the time axis is

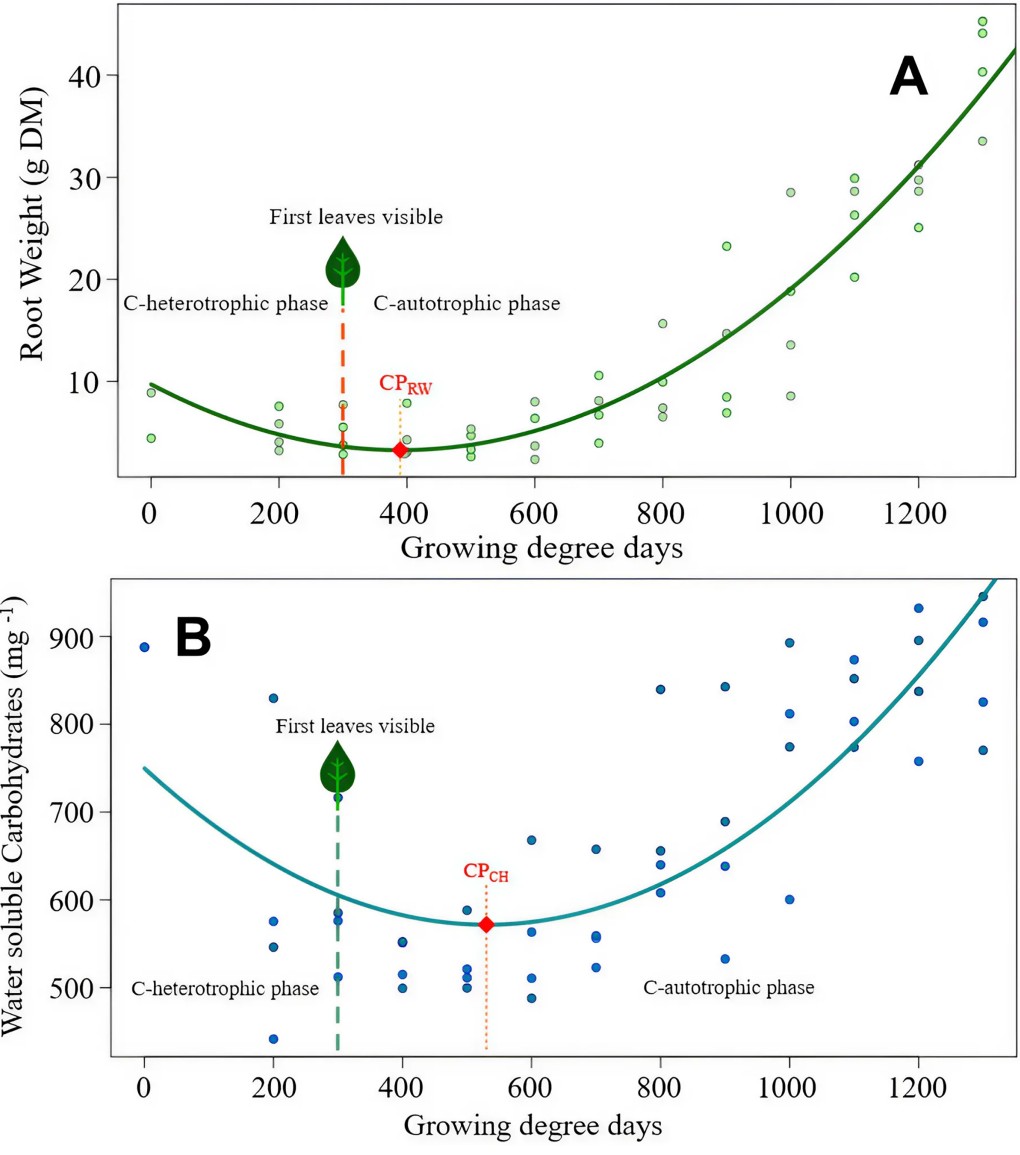

**Figure 3 *Sonchus arvensis* root reserves over time.** (A) Root weight development of *S. arvensis* as a function of experimental time. RW = 9.72 − 0.033 GDD + 0.0000424 GDD2, $R^2$ = 0.88***. Model regression line (in green) and the original data (as dots). The dashed line with a leaf at the top indicates the time of first leaf appearance, representing the shift from the C-heterotrophic to the C-autotrophic phase. The red square on the regression line marks the minimum (compensation point) according to ramet weight (CPRW) in g DM. (B) Development of RCH concentration as water soluble carbohydrates (WSC) in roots of *S. arvensis* as a function of experimental time. WSCH = 749.7 − 0.672GDD + 0.000633GDD2, $R^2$ = 0.58***. Model regression line (in turquoise) and the original data (as dots). The dashed line with a leaf at the top indicates the time of first leaf appearance, representing the shift from the C-heterotrophic to the C-autotrophic phase. The red square on the regression line marks the minimum (compensation point) according to the RCH concentration (CPCH).

relatively wide in *S. arvensis*, which also reflected in a higher residual standard error of the estimate (Table 5).

Key parameters and outcomes of the four models are compared in Table 5. Feature 'Experimental time till minimum point' equals the term 'Compensation point'. To support

**Table 5 Key parameters and output of the four regression models.**

| Thistle species | C. arvense | | S. arvensis | |
|---|---|---|---|---|
| Dependent variable | Root weight | RCH concentration | Root weight | RCH concentration |
| Predicted minimum value [Confidence Interval] | 2.75 [1.38–4.13] | 149.21 [120.08–178.35] | 3.28 [1.67–4.89] | 571.61 [533.06–610.18] |
| Experimental time until the minimum point (in GDD) | 429.5 | 818.5 | 389.5 | 530.0 |
| Duration until the appearance of the first leaf (in GDD ± SD) | 281.19 ± 62.7 | | 299.53 ± 86.47 | |
| Adjusted R squared | 0.75 | 0.73 | 0.88 | 0.58 |
| Residual standard error | 3.56 | 73.16 | 4.12 | 97.56 |

**Note:**
Root reserves as a function of experimental time.

data interpretation, measured characteristic 'Duration till appearance of the first leaf' was included in the tabulation. This duration was neither significantly affected by the kind of species (Kruskal–Wallis $\chi^2$ = 1.49, df = 1, $p$ = 0.222) nor by the root size (Kruskal–Wallis $\chi^2$ = 1.10, df = 1, $p$ = 0.296).

# DISCUSSION

Our experiments analyzed root weight, RCH concentration and RCH amount for two creeping perennial species. This is, to our knowledge, the first time that two methods of determining minimum values of root reserves were directly compared in the same experiments. We present novel insight into the reserve dynamics, which are especially lacking for *S. arvensis*.

## Species-specific root reserve dynamics

The results showed that root weight turned out to be significantly affected by the interaction 'Thistle species × Experimental time' (Table 2) although it was independent of the factor species alone. Carbohydrate concentration and amount were significantly affected by the same interaction (Tables 3, 4). These results strongly demand to evaluate the two perennial weed species individually regardless of the response variable. Hence, we could not confirm our first hypothesis about similar carbohydrate dynamics in roots of the two species. The significant interaction 'Thistle species × Experimental time' indicates differences during each species establishment growth. *Nkurunziza & Streibig (2011)* determined the carbohydrate concentrations in ramets of *C. arvense* and *Tussilago farfara* L. (propagating by rhizomes), both species being also creeping perennials from the Asteraceae plant family. The results showed different developments for the two species over time. Hence, their study also urged to analyze carbohydrate dynamics of perennial weed species individually, assuming that using two different forms of propagules (creeping roots and rhizomes) does not significantly impact carbohydrate dynamics. Although being botanical relatives and sharing the same life-form, differences in their root carbohydrate dynamics hinder transferring findings from one species to another.

While both species started with the same root weight and only slightly different carbohydrate concentrations, the development over the experimental time was drastically

different (Figs. 2, 3). At the beginning, *S. arvensis* lost root weight and carbohydrates due to resource mobilization for sprouting and early development faster than *C. arvense*. In the second half of the experiment, the RCH concentration balance of *S. arvensis* recovered much faster from the heterotrophic phase than that of *C. arvense*. The root weight and the carbohydrate concentration increased much steeper in *S. arvensis*, resulting in carbohydrate amounts of 36.4 g in all harvested roots for *S. arvensis* combined compared to 7.23 g for *C. arvense* at 1,300 GDD days. *Cirsium arvense* replenished RCH concentration rather slowly resulting in not higher values at the end of the experimental time than at the start. Together with an overall level of carbohydrate concentration never falling below 550 mg/kg of dry weight, *S. arvensis* clearly was the more efficient species in storing carbohydrates during the experimental time. We terminated the experiments in summer after the onset of flowering of both species. At that time *S. arvensis* probably filled-up most of the essential reserves. *Sonchus arvensis* is described to whither and to enter dormancy closely after seed set in early autumn (*Håkansson & Wallgren, 1972*), while *C. arvense* continues vegetative growth and thereby photosynthetic activity (*Nkurunziza, 2010*). Thus, *C. arvense* had plenty of time ahead to grow and store, while *S. arvensis* was close to finishing its seasonal growth.

## Minimum values in root reserve levels

Our results also revealed specific dynamics of root weight and RCH concentration approaching minimum values. For both species, root weights dropped earlier than RCH concentration and were rising again while RCH concentrations were still declining (Figs. 2, 3). Thus, minimum values of RCH concentration did not coincide with minimum values of root weight for both species. The factor ramet size was of no importance for these dynamics. Even though larger ramets were planted with more carbohydrate reserves due to their greater initial size, they did not rely on them longer than smaller ramets before reaching the compensation point. There was no significant impact on the RCH concentration, carbohydrate amount and subsequent reaching of the compensation point (Tables 3, 4, Figs. 2, 3) over the full experimental time for both species. Therefore, we refuse the second hypothesis that larger ramets rely on stored reserves for longer time periods than smaller ramets. Nevertheless, this hypothesis also addressed the compensation point. Obviously, there are two different compensation values, one based on root weight and the other one on carbohydrate concentration. Regardless of ramet size the compensation point for root weight occurred at 429.5 GDD for *C. arvense* and 389.5 GDD for *S. arvensis*. Compensation point based on RCH concentration resulted in a larger difference: 818.5 GDD for *C. arvense* and 530 GDD for *S. arvensis* (Table 5), hence the compensation point was reached later based on RCH concentration than on root weight. Differences in compensation point values between root weight or RCH concentration were more pronounced in *C. arvense* (389.0 GDD difference) than *S. arvensis* (140.5 GDD difference). We explain our findings with the production of new creeping roots coinciding with the minimum point in root weight for both, *C. arvense* and *S. arvensis*. In these new creeping roots the concentration of carbohydrates was lower than in older thickened roots, thereby diluting the concentration and altering the dynamic differently to that of root weight. The

production of new creeping roots probably delayed the compensation point based on RCH concentration compared to root weight. We confirm our third hypothesis that the occurrence of minimum values in root reserves depends on the root reserve determination method. Using root weight as a proxy for reserves delivers earlier minimum values than directly analyzing the carbohydrate concentrations for both species.

For *C. arvense* differences in the compensation point are evidently caused by the method of measuring. Measuring root weight resulted in the compensation point ranging between three leaves (*Verwijst et al., 2013*) and eight leaves (*Dock Gustavsson, 1997*). In our study the compensation point based on root weight laid with three leaves in this range, while favoring the results of *Verwijst et al. (2013)*. If root carbohydrate concentration (RCH) was measured, the compensation point occurred later: between eight leaves (*Nkurunziza & Streibig, 2011*) and 12 leaves (*Rodriguez et al., 2007*). With a compensation point (RCH) at 12 leaves our data confirm these results, too. Obviously, the direct measurement of RCH leads to compensation points later in the early growth of *C. arvense* sprouting from ramets.

The methodological discussion about the compensation point is exclusively served by studies on *C. arvense*, besides ours, no study on *S. arvensis* adds to this. However, in *S. arvensis* the difference between the methods of measuring is much smaller, indicating the compensation point between three (measuring root weight) and seven leaves (measuring carbohydrate concentration). The initial size of the ramets was of hardly any importance in our study. Methodologically, we used two ramet sizes with "large", doubling the weight and amount of carbohydrates of "small". Obviously, these small ramets were able to compensate for their lack of reserves at planting time. Studies on *C. arvense* (*Dock Gustavsson, 1997*) as well as *S. arvensis* (*Anbari et al., 2016b*) found more vigorous growth from larger compared to smaller ramets. However, these studies investigated ramets smaller than those we declared as "small". A significant impact appeared for ramets of approximately half the size of our "small" (*Dock Gustavsson, 1997*; *Verwijst, Tavaziva & Lundkvist, 2018*). To explain our no-effect result of ramet size we suggest that the small ramets were simply not small enough to have an impact on any evaluated response variables.

## Implications for perennial weed control

In the context of identifying a weak phase vulnerable to weed control measures in perennial weeds, the most relevant minimum value to consider would be the one evaluated by carbohydrate concentrations rather than root weight. This method provides a direct measurement of the carbohydrate content (*Wilson & Michiels, 2003*). Measuring root weight is a valuable proxy method, but might not be as accurate or specific as directly measuring the carbohydrate concentration. This distinction is important because it is not only the total amount of carbohydrates that matters but especially the concentration within the plant. Carbohydrate concentrations reflect the availability of readily usable energy resources in the plant (*Wilson, Kachman & Martin, 2001*; *Wilson & Michiels, 2003*). When the concentration of carbohydrates is low, the plant has less energy available, hence making the plant more vulnerable to weed control. In contrast, a low amount of
carbohydrates does not necessarily imply a low concentration. A plant might have lower total carbohydrate reserves but still maintain a sufficient concentration in its tissues critical for regrowth and survival. Therefore, understanding the concentration of carbohydrates gives a more accurate picture of the plant's ability to regrow after being treated.

The compensation point represents the point when root reserves are lowest and therefore, indicates an optimal time to control (*Håkansson, 2003*). The compensation point we found, one at early stages and one at delayed stages closer to flowering are both mentioned in literature to be favorable to control *C. arvense* (see reviews of *Tiley (2010)* and *Favrelière et al. (2020)*). Our findings for *C. arvense* support to control at later stages closer to flowering. Our results definitely indicate to control *S. arvensis* earlier in the growing season than *C. arvense*. For practical applications, users must be able to assign carbohydrate concentrations to the developmental stages for each of the two species investigated. Measuring the carbohydrate content directly in the field is not yet methodologically feasible.

In addition to the time of control, the method of control may vary, either belowground mechanical control or foliar application of systemic herbicides. For optimal effectiveness, systemic herbicides and mechanical disturbance may demand different growth stages. While belowground mechanical disturbance mainly requires low root reserves, herbicide applications necessitate adequate foliage as well as basipetal assimilate transport from above to belowground parts. This observation suggests that herbicide applications might need to be scheduled later compared to ploughing. However, studies on *C. arvense* indicated that basipetal transport of assimilates is already occurring at early growth stages (*Tworkoski, 1992*; *Nkurunziza & Streibig, 2011*). This can be sufficient for early herbicide application. *Tavaziva, Lundkvist & Verwijst (2019)* also stated that early herbicide spraying on *C. arvense* is as efficient as later season applications. Based on these explanation and studies, there are no clear indications of different optimal dates for using herbicides or mechanical disturbance, leading us to not distinguish between control measures.

Ramets are in practice produced by belowground mechanical control. Fragmentation of roots has already been proven to be effective in controlling *C. arvense* (*Weigel & Gerowitt, 2022*; *Weigel, Andert & Gerowitt, 2023*; *Weigel et al., 2024*). However, our results show that mechanical efforts to cut ramets from 20 to 10 cm offered no extra benefit, but rather an extra challenge as more ramets are produced. In practice, efforts to fragment roots into ramets should be evaluated for trade-offs in soil structure, soil erosion and energy consumption.

These recommendations on when and how to control are based on pot experimental results, no doubt that in fields the applied technologies of control will set own limits to this.

Until now, most references in literature focused on *C. arvense*. We strongly warn to simply transfer knowledge to *S. arvensis*. These two species differ in the root reserve dynamics over development stages. Fast and efficient storage of root carbohydrates in summer and withering aboveground in early autumn while leaving roots well fed for winter are unique traits for *S. arvensis*. Today, we would recommend to chop *S. arvensis* roots into small ramets not in autumn, but rather do that in spring. Actually, this recommendation is just based on our pot experiment, which means it is not yet fully

evidenced. The clear difference to *C. arvense* underscores the need for additional studies to better understand the growth and carbohydrate reserve dynamics of *S. arvensis* before applying generalized control strategies. Until then, recommendations for controlling *S. arvensis* should be made with caution, considering its unique carbohydrate dynamics. We strongly suggest to better research dynamics in *S. arvensis* growth and ontogenesis.

## CONCLUSIONS

Despite being botanical relatives, our results revealed significant species-specific differences in carbohydrate dynamics, with *S. arvensis* demonstrating more efficient storage and faster reserve replenishment than *C. arvense*. These findings underscore the importance of researching species individually in future studies. Our findings recommend earlier control measures for *S. arvensis* at around the seven leaf stage and later interventions for *C. arvense* around the 12 leaf stage. The faster recovery of *S. arvensis* compared to *C. arvense* likely necessitates more frequent control measures.

The root reserve determination method proved critical for identifying compensation points. Compensation points based on carbohydrate concentration occurred later than those based on root weight measurements for both species, especially for *C. arvense*. Our research highlights the importance of carbohydrate concentration measurements, as it is a more precise indicator for determining the timing of thistle control measures compared to root weight.

The initial ramet size did not significantly affect both species' root reserve dynamics within the tested size range. As a consequence of this limited effect of fragmentation, practical applications to fragment roots into small ramets through belowground mechanical control must be evaluated for trade-offs in soil structure, soil erosion, and energy consumption. While ramet size had no significant effect on root weight or carbohydrate concentration in this study, further research with smaller initial sizes may reveal size-dependent dynamics.

Future research should validate these findings under field conditions, with particular attention to *S. arvensis*'s unique traits, such as its rapid carbohydrate storage and early withering.

## ACKNOWLEDGEMENTS

We sincerely thank our colleagues Rosa Minderlen, Diana Sicard and Ingolf Gliege in the Crop Health group at the University of Rostock for their technical assistance.

### Funding

This project has received funding from the European Union's Horizon 2020 research and innovation programme under grant agreement No 771134. The project AC/DC weeds was carried out under the ERA-NET Cofund SusCrop (Grant N°771134) in the Joint Programming Initiative on Agriculture, Food Security and Climate Change (FACCE-JPI). The German contribution is funded by DFG (Deutsche Forschungsgemeinschaft), GE 558/

3-1. There was no additional external funding received for this study. The funders had no role in study design, data collection and analysis, decision to publish, or preparation of the manuscript.

## Grant Disclosures

The following grant information was disclosed by the authors:
European Union's Horizon 2020 Research and Innovation: 771134.
AC/DC Weeds was Carried out under the ERA-NET Cofund SusCrop: N°771134.
Joint Programming Initiative on Agriculture.
Food Security and Climate Change (FACCE-JPI).
DFG (Deutsche Forschungsgemeinschaft): GE 558/3-1.

## Competing Interests

The authors declare that they have no competing interests.

## Author Contributions

- Marian Malte Weigel conceived and designed the experiments, performed the experiments, prepared figures and/or tables, authored or reviewed drafts of the article, and approved the final draft.
- Sabine Andert conceived and designed the experiments, authored or reviewed drafts of the article, and approved the final draft.
- Manuela Alt performed the experiments, authored or reviewed drafts of the article, and approved the final draft.
- Kirsten Weiß performed the experiments, authored or reviewed drafts of the article, and approved the final draft.
- Jürgen Müller analyzed the data, prepared figures and/or tables, authored or reviewed drafts of the article, and approved the final draft.
- Bärbel Gerowitt conceived and designed the experiments, authored or reviewed drafts of the article, and approved the final draft.

## Data Availability

The raw data is available in the Supplemental File.

## Supplemental Information

Supplemental information for this article can be found online at http://dx.doi.org/10.7717/peerj.19155#supplemental-information.

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
