# Peer review of "Root fragment weight and carbohydrate dynamics of two weedy thistles Cirsium arvense (L.) Scop. and Sonchus arvensis L. during sprouting"

_PeerJ, doi:10.7717/peerj.19155_

## Round 0.1 · original submission · Minor Revisions

The dynamics of carbohydrates in perennial weed species play a significant role in managing these weeds in various crop fields. I thoroughly enjoyed reading your article and appreciate your efforts, as a fellow weed scientist, to shed light on this important subject. However, it is essential to address certain technical details to enhance the article further. I encourage you to carefully review the reviewers' suggestions and thoughtfully consider each recommendation. If you find yourself in disagreement with any specific suggestions, providing a clear and well-supported rationale for your viewpoint would be highly beneficial.

Reviewer 1 ·

Basic reporting

Presented in the Additional Comments

Experimental design

Presented in the Additional Comments

Validity of the findings

Presented in the Additional Comments

Additional comments

Final Review
I consider it to be a relevant contribution to the study of carbohydrate dynamics in perennial species. The work is technically sound and well-structured. However, some minor revisions are necessary to enhance clarity, contextualization, and the presentation of results.
Points for Revision and Locations
1. Methodology - Justification of the chosen methodology (page 8): In the section "Statistical analysis", the choice of the second-degree polynomial model could be better justified. I suggest adding a brief explanation as to why it was deemed the most appropriate for this study.
2. Results and Discussion - Practical interpretation (pages 13–14): In the subsection "Implications for perennial weed control", include a clearer discussion on how the results can be applied to agricultural management practices, particularly regarding the use of carbohydrate metrics for weed control.
S. arvensis data (page 12): In the section "Species-specific root reserve dynamics", the results related to S. arvensis could be explored in greater detail to reinforce the practical relevance of the study, given the limited information available on this species.
3. Figures and Tables - Figures (pages 18–20): The legends for Figures 2 and 3 could be expanded to explain the significance of the inflection points in the graphs and their relationship to critical moments in the experiment.
4. Writing - Structuring hypotheses (page 7): Reorganize the "We hypothesize:" section by separating each hypothesis with clear and objective explanations.
Recommendation
Acceptance with Minor Revisions. The manuscript is solid and presents significant contributions. The suggested revisions aim to improve the clarity and applicability of the results without altering the substance of the work.

Reviewer 2 ·

Basic reporting

Raw data are shared, however I couldn't find discussed data about number of leaves at the compensation point for this study there.

Experimental design

Methods part needs some elucidation. See more in additional comments.

Validity of the findings

I don't understand well the rationale behind refusing the second hypothesis. The explanation could be expanded.

Additional comments

Lines 56-57. In other literature, the terms used for root system of Cirsium arvense are vertical and horizontal roots. You can keep your terms, as creeping root are clear for me, but in case of common roots, you should better describe, what is their definition.
Lines 111-112 How was treated the original fragment? Was it considered to be a part of the reserves?
Line 142 – I am missing here information on how the pots were placed in the greenhouse. Were they organised following some design, randomly or in blocks?
Line 177 – What are exactly the nutritious roots? Can you provide an explanation?
Lines 233-237 – When you mention ramet size, it is not clear, which parameter you are talking about. In Table 1, reader can find small/large or diameter and length.
Lines 297-300 – I wonder, whether it would be possible to present visually some graph showing carbohydrate content dynamics in re-sprouting process. And how much it is either similar of dissimilar in comparison to root weight curve in fig. 2 and fig. 3. Such comparison could tell us to which extent is the weight of root proxy of its carbohydrate content. I could find carbohydrate content results only in Table 4.
Line 310 – I would better call the process establishment growth, not ontogenesis. You work within very short period of time to cover whole ontogenesis.
Line 311 - Tussilago. Farfara – change to Tussilago farfara
Line 312 – ‘Please visit…’ is awkward phrasing. Is the method so different for discussion of results?
Line 315 – They are both creeping weeds, if you consider it the same life-form, however morphologically rhizome is belowground stem, so we may expect different intrinsic processes behind carbon dynamics during re-sprouting.
Line 320 - ‘lost root weight’ – what do you expect to be the mechanism behind loosing root weight at the heterotrophic phase. Did you observe some shedding of dead parts of roots?
Line 332 – ‘close to finish’ of what? Ending its seasonal growth or close to death?
Line 339 – Factor ramet size was significant in root weight analysis, no? So, I don’t see so clearly the point to refuse second hypothesis based on that. May be, you should just explain better the rationale behind.
Lines 358-366 – Where are the data about number of leaves and compensation point from your study? I couldn’t find them in results nor in your raw data.
Line 357 – the concentration of carbohydrates
Lines 371-373 – This is confusing for me. I can’t see ‘the occurrence of the compensation point’ in the table 3 and 4. Is it connected to ‘experimental time’, which has however significant effect in case of carbohydrate amount in table 4. Can you elucidate this part?
Lines 385-388 – I consider this paragraph unnecessary, as it doesn’t bring more information.

Reviewer 3 ·

Basic reporting

1. In the discussion section, line 377, Anbari et al.2016 and line 392, Wilson,2003 are not included in the reference section.
2. The references in the article "Anbari S, Lundkvist A, Forkman J, Verwijst T (2016b) Population dynamics and nitrogen allocation of Sonchus arvensis L. in relation to initial root size. Acta Agriculturae Scandinavica, Section B — Soil & Plant Science 66:75–84. https://doi.org/10.1080/09064710.2015.1064540"; are not included in the article.

Experimental design

no comment

Validity of the findings

no comment

Additional comments

The article is very well prepared in terms of fiction, hypothesis, conclusion and discussion. I wish you continued success.

---

## Round 0.2 · accepted · Accept

I would like to thank you for accepting the referees' suggestions and improving your article based on their suggestions. Your article is ready to publish. We look forward to your next article.

The Section Editor made the following suggestions that you might consider during the proofing stage:

> I believe adding "two weedy thistles" to the title could attract broader readership: "Root fragment weight and carbohydrate dynamics of two weedy thistles (Cirsium arvense (L.) Scop. and Sonchus arvensis L.) during sprouting"

> line 238 - I would replace "The results allow to model" with "These data allow modeling of"

Reviewer 3 ·

Basic reporting

no comment

Experimental design

no comment

Validity of the findings

no comment'

Additional comments

The suggested corrections have been made. I wish you continued success in your work.